# System Dynamics Modeling in Local Water Management: Assessing Strategies for the City of Boerne, Texas

**Vianey Rueda [1,†], Michael H. Young [2,*], Kasey Faust [3], Ashraf Rateb [2] and Benjamin D. Leibowicz [4]**

1   School for Environment and Sustainability, University of Michigan, Ann Arbor, MI 48109, USA
2   Bureau of Economic Geology, Jackson School of Geosciences, The University of Texas at Austin, Austin, TX 78758, USA
3   Department of Civil, Architectural, and Environmental Engineering, Cockrell School of Engineering, The University of Texas at Austin, Austin, TX 78712, USA
4   Walker Department of Mechanical Engineering, Cockrell School of Engineering, The University of Texas at Austin, Austin, TX 78712, USA
*   Correspondence: michael.young@beg.utexas.edu
†   Research conducted while at: Energy and Earth Resources Graduate Program, Jackson School of Geosciences, The University of Texas at Austin, Austin, TX 78712, USA.

**Abstract:** As more pressure is exerted onto water sources, hydrologic systems may be altered in ways that are difficult to predict. In Texas, water deficits can become widespread as sources are strained beyond capacity. For smaller communities, such as Boerne, Texas, water management and planning is a way to prepare. The supply-demand water balance in Boerne is conceptualized through causal loop diagrams and system dynamics modeling. Through stakeholder engagement, xeriscaping, rainwater harvesting, and smart meters were chosen as interventions, each varied in adoption levels. The resulting 125 combinations were analyzed under three scenarios: a base case assuming maximum supply of water is firm, and two responses to a meteorological drought. Results show that the city can effectively forestall a deficit. Different combinations of adoptions can achieve the same goal, giving the city optionality in choosing strategies that are best suited for its needs and constraints. Rainwater harvesting was found to be the dominant intervention influencing demand, but its influence is reduced in the two drought scenarios. Xeriscaping was the second most influential intervention and smart meters for irrigation had no effect on demand. The approach used in this study highlights the interdependency between community adoption of conservation strategies and the importance of considering these relationships using systems modeling.

**Keywords:** system dynamics; water management; Boerne; Texas; stakeholder engagement; sustainability and resilience

## 1. Introduction

Future accessibility, quality, and quantity of water resources are threatened from the growing pressure of population, economic development, and climate change [1]. Once relatively balanced, hydrologic systems have been and will continue to be altered by competing stresses, generating changes that are difficult to predict and quantify, especially at local scales. Scale-appropriate analyses can reveal solutions feasible for a given city by understanding their relationship with water and how these may vary given local circumstances. While national- or state-scale water management and planning approaches can prove too abstract, vis-à-vis system components and stakeholders, the municipal/district level facilitates identification of relevant stakeholders and system boundaries. This scale of analysis is also naturally bounded by service areas and jurisdictional boundaries.

Sustainable and resilient water management and planning will prepare Boerne, Texas for water shortages that are likely to occur in the future. In 2015, the city contracted with HDR Engineering Inc. [2] to guide a water planning process to account for rapid population

growth. The plan projects water shortages based on two sources of data, population estimates from the city itself and population estimates from the Texas Water Development Board (TWDB) in the 2016 Region L Water Plan [3]. This study highlights the importance of projected population growth, changes in per capita consumption, and the use of treated wastewater. With city data, shortages were estimated by 2070, while TWDB data estimated shortages by 2050 and worsening through 2070. While population estimates from the TWDB are smaller than those of the City of Boerne, estimated consumption by the TWDB is greater at 0.73 m$^3$ (192 gallons) per capita per day (PCD) in 2020 to 0.71 m$^3$ (187 gallons) PCD in 2070. Furthermore, the city estimates reclaimed water use to increase to almost $2.5 \times 10^6$ m$^3$ (2000 AF) by 2070, while the TWDB found this to be a constant $8.6 \times 10^3$ m$^3$ (7 AF).

With a population of 17,250 as of 2020, Boerne, Texas typifies a relatively small community adjacent to a large municipal city (i.e., San Antonio). To increase the sustainability and resilience of its water system, Boerne—and other communities that are also undergoing and managing significant population growth—must address questions related to their water resources: what types of policies should the city pursue, if any? Which conservation strategies are most effective in reducing water demand and how would they fare under drought conditions? How can the city avoid a water deficit?

Answering these questions at present increases the adaptability of a city to future circumstances. Alternatively, delaying action can result in reactive measures during some future drought scenario. These issues become more salient as community populations grow, and the difference diminishes between supply—which has a physical upper limit—and demand—which is closely tied to decisions made within the community and is affected by growth in surrounding areas. In the Texas Hill Country, water supply and infrastructure are closely tied and limited; the threat of inter- and intra-city competition for limited resources will continue to be an issue in areas where both growth and potential resource shortages intersect.

This research will explore the effectiveness of various options and strategies to be tested, without the lag time (or risk) of experimenting on real systems. Refsgaard et al. [4] highlights the interaction between water management and modeling processes and the methodologies to assess uncertainty, including scenario analyses. Figure 1 shows the interaction between relevant stakeholder(s) (on the left) and the modeling step they helped inform (on the right). We incorporate scenario analysis and the constant feedback of various stakeholder groups from Boerne, Texas throughout the modeling process.

In this study, we use Causal Loop Diagrams (CLDs) and System Dynamics modeling for reasons described below. CLDs represent an individual hypothesis of how a system functions and is the first step in framing the scope of a System Dynamics (SD) model [5]. CLDs map causal relationships and identify feedback loops within a process or system [6]. In a CLD, the polarity of each arrow highlights the reinforcing or mitigating relationships that the arrow connects. The overall loop sign is obtained by multiplying the signs of the variables involved in the feedback [7].

Since its conception in the 1950s [8] and with further development [9], SD has been applied to a number of topics specific to water, such as illustrating the effects of proposed strategies while raising stakeholder awareness of resource problems [10]. The appropriateness of SD modeling techniques to address water management concerns specifically was shown by [11–13]. Elsawah et al. [11] found that SD modeling can be applied to water allocation problems at various scales, with a tendency to use this type of modeling for decision-making and social education. Additionally, Winz et al. [12] found that, because it requires explicit acknowledgement of assumptions and identification of uncertainties, SD modeling represents a transparent method that can confidently inform policy recommendations. Finally, Karimlou et al. [13] reasoned that SD can help choose the most efficient management strategies by helping managers observe the linked changes occurring in the system.

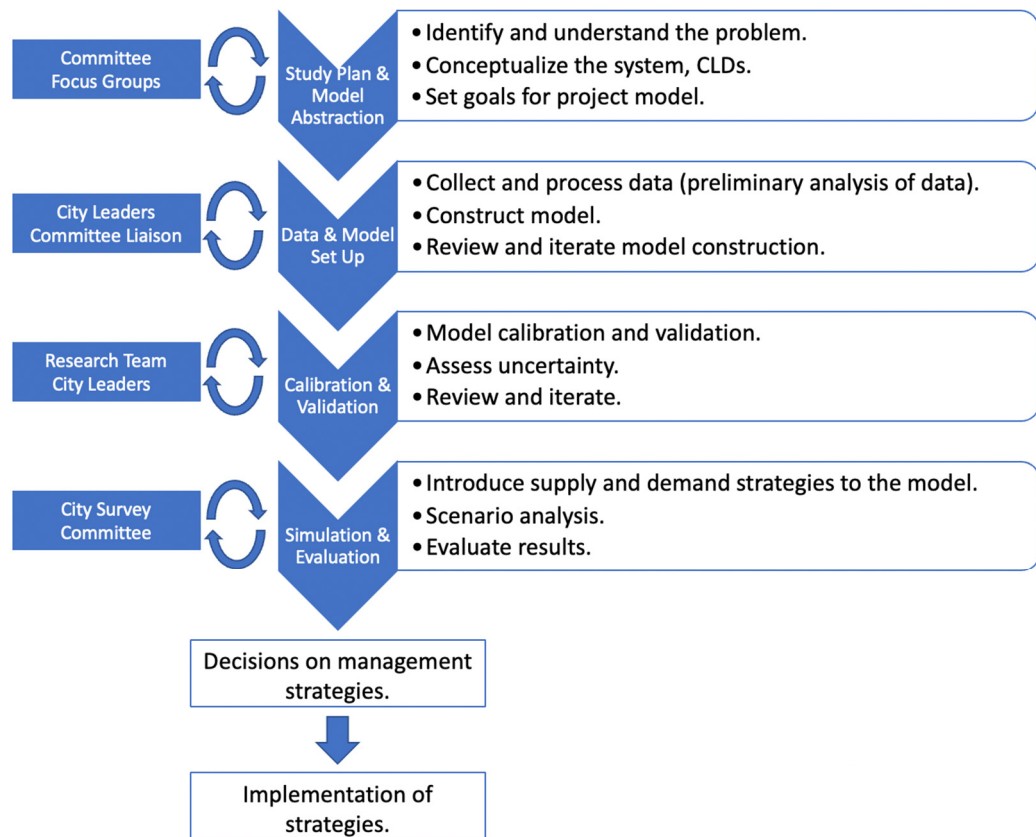

**Figure 1.** Interactions Between Stakeholders and the Modeling Process.

Stave [14] explains that SD emphasizes finding the *causes* of problematic behavior, assisting investigators and stakeholders as they propose solutions. This is important because one difficulty of resource management is the inability to comprehend or foresee the cascading effects of changes made to the system due to a lack of understanding of how system components are interconnected. Martone et al. [15] explains similar observations. Newell et al. [16] further this argument, explaining that linear thinking suggests that doubling a cause doubles the effect, when in reality causation in complex systems is impacted by feedback. The usefulness of SD is enhanced by feedback analyses, which have the capacity to visually demonstrate how changes in some elements affect other elements and the overall dynamics of the system [16–19].

While the above literature indicates the usefulness of SD modeling in water-related fields, to our knowledge, we are unaware of studies dedicated specifically to understanding the effectiveness of strategies in reducing sector-demand at a municipal/city scale, while also incorporating stakeholder participation throughout the modeling process, distinguishing demand from different sectors, and incorporating different adoption levels in strategy combinations. The five case studies used by Karimlou et al. [11], for example, are not local scale, but instead include a metropolitan region, groundwater systems, and a river basin. Stave [10], on the other hand, uses SD modeling to assist stakeholders in Las Vegas, Nevada to understand management options based on their capacity to reduce demand, but the produced model incorporates all demand as either indoor or outdoor. Unlike Winz et al. and Karimlou et al. [10,11], this research breaks indoor and outdoor demand further into residential, municipal, and business sectors to generate a greater understanding of the effect of each sector on the local system. Altogether, this research incorporates SD modeling, demand-reducing interventions, stakeholder participation, and a more detailed understanding of demand, at the municipal level.

The chosen approach was tailored based on local knowledge and direct feedback by utility managers and stakeholders. The value of stakeholder participation and bottom-up approaches to water management and planning is exemplified in the Texas State Water Plan [20] as well as in the literature [21–29]. As part of the socio-hydrological system, Boerne residents are as important and relevant as the economic, political, and scientific aspects of water management and planning. Residents demand the majority of water entering the city, and they create an important leverage point in the system where conservation strategies can be introduced; through their support or opposition, residents can influence the success or failure of these actions.

The overall goal of this research is to demonstrate the usefulness of community engagement and SD modeling in creating water management plans. We use the community of Boerne, Texas as the case study, though we note that this workflow and community engagement outcomes can be applied in other cities focusing on improving sustainability and resilience of their water systems. Effectiveness of a policy here is measured based on the ability of an intervention to delay a deficit in water supply. Through stakeholder engagement, other communities can incorporate data specific to their situation.

## 2. Materials and Methods

### 2.1. Study Area

Located in the south of Kendall County, the City of Boerne is a medium-sized community in the San Antonio MSA and is surrounded by the Texas Hill Country. Cibolo Creek, an important component of the water distribution system (WDS) in Boerne, is a tributary of the San Antonio River and flows through the city, feeding into Boerne City Lake. With respect to water, the city lies within many important geographical and governmental boundaries, being within the jurisdiction of the Cow Creek Groundwater Conservation District (GCD), the 9th Groundwater Management Area (GMA), and the Region L Planning Group. Climate in the area around Boerne is humid subtropical to semi-arid [30], with an average annual temperature between 15.6–18.3 °C (60–65° F) and less than 88.9 cm (35 in.) of precipitation per year [30]. High summer temperatures and low precipitation make the region prone to moderate to severe droughts.

The city obtains its water supply from surface and subsurface sources. Boerne obtains ~33% of its water from the Trinity Group Aquifers. Boerne City Lake supplies 25% and the remaining 42% comes from Canyon Lake through a contract with the Guadalupe Blanco River Authority (GBRA) [31]. Specific yearly permitted allocations can be found in Table 1. By the end of the 2020 calendar year, the city had only purchased approximately 36% of its total contract with GBRA and has been purchasing below their GBRA contract supply for the last ten years [32] (Figure S1 in Supplementary Materials). This is because demand has not reached levels of consumption necessary of greater purchases, and because the City of Boerne has maintained a very proactive water resources planning programming, especially given the current and expected growth in the region and the vulnerability of the community to droughts. The future water supply from GBRA will depend on water availability and possible policy decisions regarding curtailment.

**Table 1.** Water Supply by Source.

| Source | Amount ($m^3$/year) |
|---|---|
| Boerne Lake | 1,027,488.5 (833 AF *) |
| Trinity Wells | 1,233,479.6–2,281,937.3 (1000–1850 AF *) |
| Canyon Reservoir | 4,454,095 (3611 AF *) |

* 1 AF = 325,851 gal.

Efforts to increase supply of reclaimed water have also increased. Currently, the city operates the Esser Road Wastewater Treatment Plant, with a capacity of 4542 m$^3$ (1.2 MGD), and the Old San Antonio Road Wastewater Treatment and Recycling Center, with a capacity of 5300 m$^3$ (1.4 MGD). The vast majority of reclaimed water from Esser Road is used for maintaining streamflow in Cibolo Creek. The majority of reclaimed water from the Old San Antonio Road facility is used for outdoor irrigation in residential areas, city parks, and other public areas [2]. From October 2019 to October 2020, reclaimed water accounted for approximately 11% of total water used by the city. Through consultation with the Utilities Director, we decided that water reclamation was best classified as a conservation strategy. We reasoned that wastewater reclamation is not an increase in supply that can be used for indoor consumption. Rather, its use reduces pressure on potable water supply that would otherwise be used for outdoor irrigation. Therefore, increased use of reclaimed water represents a conservation strategy for reducing potable water use.

### 2.2. Focus Groups and Community Survey

As Figure 1 indicates, the workflow includes focus group sessions with Boerne residents and a community-wide distributed survey. The sessions and survey were intended to provide information and context for understanding community perception of supply and demand of their municipal water system. Briefly here (more details in Appendix A), focus group sessions were organized around four specific stakeholder groups (municipal and county agencies, business leaders, community service organizations, and environmental groups). Members were identified through local networks, but with a goal of ensuring a diversity of backgrounds, perspectives, and influence. During focus group sessions, participants discussed their understanding of system components and existing relationships between these components, eventually yielding a CLD. Through a survey (described in Appendix A) distributed throughout the community using print and social media, respondents (n = 324) identified conservation strategies from a list created by project participants they would be more willing to adopt. The strategies, prioritized by the community, were used in the systems dynamic model.

### 2.3. Causal Loop Diagrams and System Dynamics Modeling

A preliminary CLD with knowledge of the water system in Boerne was created and vetted through several meetings with different stakeholders, including the city utility office. Given their knowledge, different stakeholder groups informed various parts of the CLD, including subject matter experts such as local water purveyors and hydrologists, as well as focus group sessions. The final CLD (Figure 2) was completed after several meetings with the Boerne Utility Director. We note that agriculture is not present in the city, nor is heavy industry with intensive water needs (i.e., manufacturing is limited to businesses in the service sector).

The CLD illustrates the water system in Boerne and informed the stocks, flows, parameters, and variables in the SD model (The AnyLogic Company, version 8.7.5; Figure 3). Descriptions and values for each of the variables were generalized to facilitate use of the model by other interested parties (reference Table S1 in Supplementary Materials for a description and values of model variables). First, we focused on supply and demand. Water source was conceptualized as the sum of three water sources (WS) (labeled WS1, WS2, and WS3) (Figure 3). The magnitude of each source is the maximum amount outlined in either a contract or permit (Table 1). The sources are pooled together and are triggered to flow into a stock on January 1 of every year. Water is withdrawn from this stock by indoor and outdoor demands on a daily basis. At the end of the year, water not withdrawn is accounted for and removed from the stock in order to prevent accumulation of unused water.

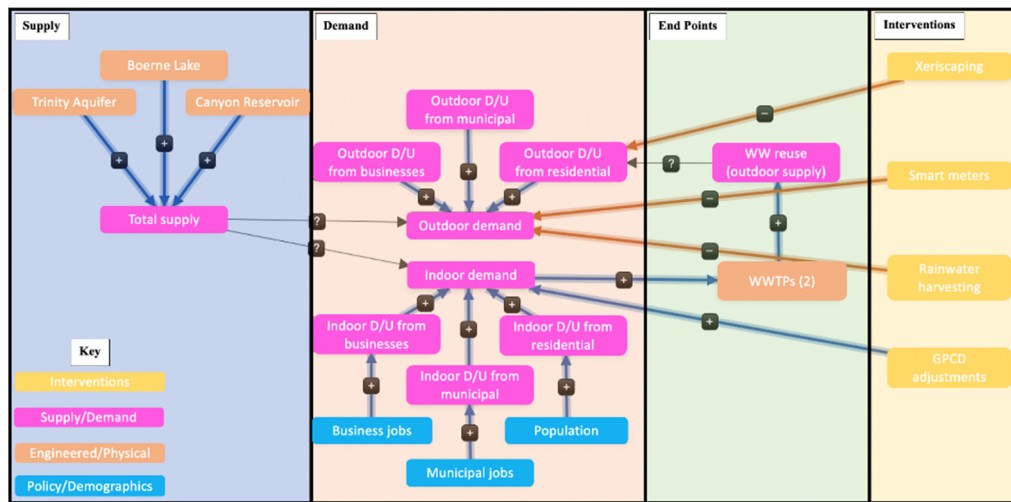

**Figure 2.** Water System Causal Loop Diagram for Boerne, Texas. Note: Demand/Use (D/U); Wastewater Treatment Plants (WWTPs). The mentioned conservation practices include smart meters, xeriscaping, and rainwater harvesting.

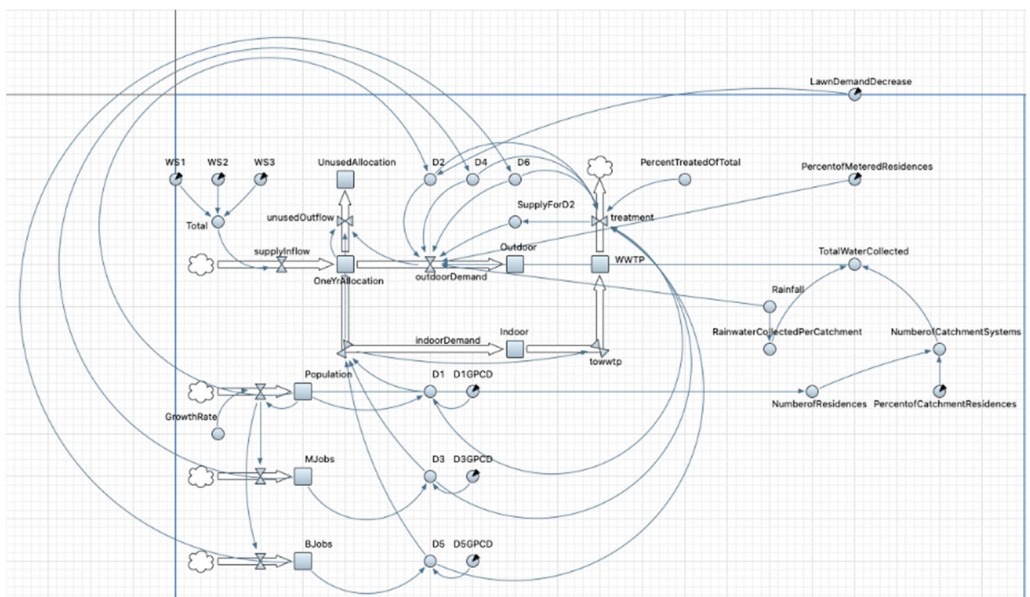

**Figure 3.** System Dynamics Model of the Water System in Boerne, Texas.

The decision to pool water into a stock that would be released on January 1 is based on the overall goal of the model— to assess broader benefits of specific interventions on long-term supply-demand dynamics of the system. This approach would be more illustrative than using daily supply dynamics given that trying to simulate the dynamics of community member decision-making on their daily water consumption (e.g., will they irrigate the day after precipitation?) would over-reach the abilities of the model. This would also make calculations and extrapolations for an approximately 50-year time period difficult and likely ambiguous.

Indoor and outdoor water uses are each calculated by their respective demands (D)—D1, D3, and D5 for indoor, and D2, D4, and D6 for outdoor. Demand for each is calculated based on the number of people for a given population type (e.g., residential, municipal, or business) multiplied by the respective per capita water use. Units will be in $m^3$/capita/day (PCD), followed by gallons/capita/day (GPCD) in parenthesis. Water for outdoor irrigation is assumed to exit the system via evapotranspiration (i.e., we assume

no shallow recharge to groundwater or surface water), while water for indoor purposes flows to a wastewater treatment plant. Once treated, a portion of this water flows back into the system to offset demand from residential outdoor irrigation (i.e., D2), and another portion is released to Cibolo Creek, where it exits the system. Model time units are in days and the simulation runs from 1 January 2022 through 31 December 2070. The model simulation begins on January 1 because this is when the yearly water supply enters the system. Discussion of modeled interventions is in Section 2.7.

Labels in the SD model were intentionally generalized so that other communities would need only update values, rather than redesign the model. For example, a water source is labeled as WS instead of the specific name of that water source since these names can change across communities.

*2.4. Incorporating Community Feedback*

Community engagement outcomes from Boerne residents were incorporated as feedback throughout this research. Focus group sessions and a publicly released survey are part of the [33] project in Boerne and thus conducted outside of this specific research; more information on these can be found in the Supplemental Information. Community engagement outcomes were relevant to this research, as participants provided their perspectives of the system, and subsequently informed Figure 2. The top three most popular strategies obtained from these community engagement efforts were introduced into the model as interventions and assessed in terms of their effectiveness in delaying a deficit in supply. Only the top three strategies were chosen to keep the modeling tractable. These strategies were tested through scenarios in which each strategy was adopted at various levels separately and in combination.

*2.5. Model Assumptions*

Data provided by the City of Boerne were available at daily increments, including water supply by source—including reclaimed wastewater—beginning 19 May 2000 to date. The calibration period for the model is from 1 June 2016 through 26 February 2021. This period was chosen because it spans the time period for which all water source data were available. Information with respect to seasonal patterns, indoor versus outdoor differences, and management of wastewater treatment was necessary to inform the model on the volume of water delivered to each sector throughout the 50-year modeling period. Educated assumptions were co-created with representatives from Boerne Utilities, including:

- Differentiation between residential and non-residential water use represented by an 80%–20% breakdown. This initial breakdown of water use between residential and nonresidential sectors is an educated assumption based on local expert knowledge. Recording actual data on this breakdown could build confident in decisions about water management and economic growth moving forward. This breakdown can be adjusted depending on specific municipal conditions.
- Outdoor water consumption is greatly impacted by seasonality (e.g., landscape irrigation). Seasons were adjusted to fit the transition months used by the city. Transition months are the periods between each season and can be used to adjust seasonality given geographic location.
  - Summer, 15 May–14 October; 60–40% indoor-outdoor
  - Fall, 15 October-30 November: indoor use increases linearly from 60% to 90% while outdoor use decreases linearly from 40% to 10%.
  - Winter, 1 December–28/29 February; 90–10% indoor-outdoor
  - Spring, 1 March–14 May: indoor use decreases linearly from 90% to 60% while outdoor use increases linearly from 10% to 40%.
- Treated wastewater supply will always exceed demand because only new construction (homes, businesses) would be added to piping infrastructure. With the guidance of the Utilities Department of Boerne, this allowed us to simplify the volume of flow from indoor demand to the wastewater treatment plants to be 50% of total indoor demand.

- Volume of water treated and returned to the system is based on the percentage of total demand. Future projections of this percentage are based on expected housing expansion that will be connected to appropriate piping. We assumed this percentage to be within 10–15% of total water use. As of 2020, treated wastewater was 11% of total water use. Beyond 2020, this percentage is assumed to grow by one percentage point per year until 15% is reached and will remain at 15% until the end of the simulation period.
- Ratio of residential population to municipal employees is assumed to be 63:1. These demographic ratios were coupled to proportion of total demand of each sector to obtain separate per capita consumption values for sector-specific indoor and outdoor uses.
- Indoor use does not fluctuate season to season. Given that little is known or recorded about variability of indoor consumption rates, we assume that residential use remains constant from season to season. Understanding the impacts of weather, comfort, and cost of water on altering indoor consumption patterns would improve the accuracy of the model.

### 2.6. Verification and Validation

Model validation was done using data from the four-year period described above. Simulated total demand—including demand across all sectors for indoor and outdoor—captures the seasonality of observed total demand (Figure 4), increasing and decreasing during respective shoulder months. Precipitation is included to show its influence on demand, particularly during shoulder months. For example, in late 2018 and mid 2019 (Figure 4), a series of rainy days reduced demand (mainly from outdoor irrigation). The average difference between known demand and modeled demand varies season to season, year to year, and sector to sector. We did not investigate the specific relationship between precipitation rates and user behavior.

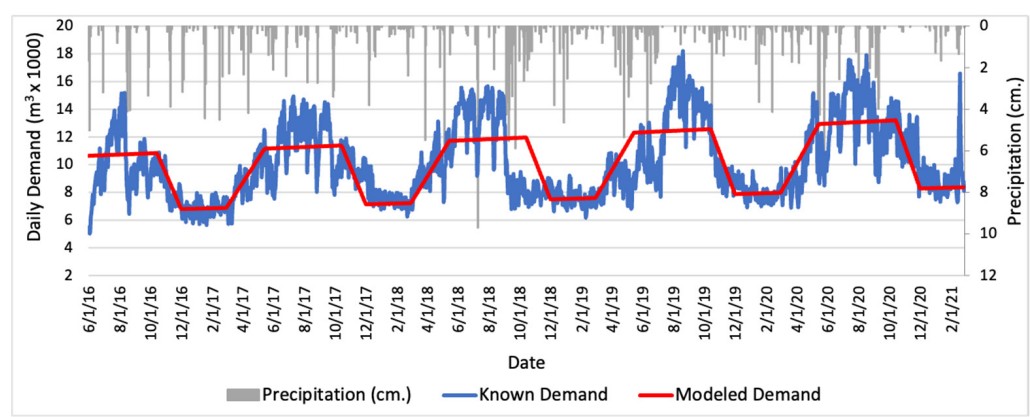

**Figure 4.** Calibration Period for the System Dynamics Model (1 June 2016–26 February 2021).

To evaluate model performance, we used quantitative and qualitative metrics for verification and validation. The Nash-Sutcliffe model efficiency coefficient [34] is widely used in hydrologic modeling as an indicator of performance. Depending on the context, different values can be interpreted as having a good fit. For example, SWAT [35] and Moriasi et al. [36] state that a value of 0.5 and above is a satisfactory fit, while Christiansen [37] noted that a 0.5 value is a good fit.

Table 2 lists the NSE value obtained for this model, as well as values of other statistical indices such as root mean square error and mean bias error. The NSE value obtained for the observed dataset is 0.54 (Table 2a), surpassing the 0.5 threshold. Table 2b lists the values of these metrics for increasing values of per capita outdoor consumption rates for all seasons of the observed dataset. These values indicate that statistical indices become worse off as outdoor per capita consumption rates increase, thus the model maintained base values. We note that maximum yearly allocation of water is over $6.44 \times 10^6$ m$^3$ (1.7 billion gallons) and

unaccounted water is 10–12% of total daily production. These values are important because they highlight areas of uncertainty within the system. Overall, the model is able to capture system behavior while not accounting for error given precipitation and unmetered water.

**Table 2.** Metrics to Measure Predictive Ability of the Model. (**a**) Error Metrics for Dataset. (**b**) Error Metrics while Varying Residential Outdoor Per Capita Demand Values.

| (a) | |
|---|---|
| **Metric** | **Value** |
| Mean Absolute Error (MAE) [1] | 1383.2 |
| Root Mean Square Error (RMSE) [1] | 1832.5 |
| Mean Bias Error (MBE) [1] | 50.8 |
| Mean Absolute Percentage Error (MAPE) [2] | 13.6 |
| Nash-Sutcliffe model efficiency coefficient (NSE) [3] | 0.54 |

| (b) | | | | | | |
|---|---|---|---|---|---|---|
| | $0.24 \, m^3$ PCD (63 GPCD) | $0.25 \, m^3$ PCD (65 GPCD) | $0.26 \, m^3$ PCD (70 GPCD) | $0.28 \, m^3$ PCD (75 GPCD) | $0.3 \, m^3$ PCD (80 GPCD) | $0.32 \, m^3$ PCD (85 GPCD) | $0.34 \, m^3$ PCD (90 GPCD) |
| N | 1732 | 1732 | 1732 | 1732 | 1732 | 1732 | 1732 |
| MAD or MAE [1] | 1383.195 | 1386.315 | 1422.040 | 1494.895 | 1614.257 | 1784.053 | 1994.118 |
| MSE [1] | $8.87 \times 10^8$ | $8.93 \times 10^8$ | $9.63 \times 10^8$ | $1.11 \times 10^9$ | $1.34 \times 10^9$ | $1.64 \times 10^9$ | $2.02 \times 10^9$ |
| RMSE [1] | 1832.535 | 1838.920 | 1909.559 | 2050.783 | 2249.272 | 2491.444 | 2765.785 |
| MBE [1] | 50.797 | (72.100) | (379.418) | (686.768) | (994.068) | (1301.371) | (1608.701) |
| MAPE [2] | 13.593 | 13.817 | 14.613 | 15.713 | 17.178 | 19.022 | 21.155 |
| NSE [3] | 0.543 | 0.540 | 0.504 | 0.427 | 0.311 | 0.155 | (0.041) |

[1] Error metric expressed in units of observations, $m^3$. [2] Error metric expressed as a percentage. [3] Error metric expressed as a range from -inf. to 1.

Furthermore, Sargent [38] describes that when a research team is small in size, the best approach is for model users to decide alongside developers the validity of the model. For this more qualitative method, results of the model were shared with utility officials, and after several meetings, the output from the model was deemed acceptable.

## 2.7. Parameter Variation

The top three community-chosen interventions were varied across five levels of adoption, generating 125 different combinations (Table 3). All combinations were compared with current consumption rates: total residential consumption of $0.59 \, m^3$ (155 gallons) PCD, broken down by $0.35 \, m^3$ (92 gallons) indoor and $0.24 \, m^3$ (63 gallons) outdoor. This is the base output (0% adoption for each intervention), represented by $X_1 R_1 M_1$.

Xeriscaping (intervention X) eliminates the need for irrigation completely and is incorporated through the resulting water saving potential. Water saving potential is the product of the difference of water consumption per day between the two landscapes (we assume St. Augustine as a representative turfgrass for central Texas), an average lawn size of 701.6 sq. m. (7552 sq. ft.) (subtracting the average roof size from the average lot size), an assumed 20% adoption in residential homes, and 0–100% conversion of lawn space from turfgrass to xeriscaping; 20% adoption is based on the demographic makeup of Boerne.

Rainwater harvesting for landscape water needs (intervention R) is varied according to percent of households that install a rainwater harvesting system. Per Krishna et al. (2005), rainwater harvesting potential is the product of the following:

$$rainfall \; depth \times roof \; area \times 0.85 \; collection \; efficiency \qquad (1)$$

Smart metering (intervention M) also varies according to the number of participating households. When installed, irrigation is not used when precipitation is equal to or greater than turfgrass water needs.

**Table 3.** Ranges and Intervals of Parameter Variation.

| Strategy (Top 3) | Identification and Range | Options | Increment |
|---|---|---|---|
| X: Xeriscaping | X1 *: 0%, X2: 5% X3: 10% X4: 15% X5: 20% of water saved | 5 | 5% |
| R: Harvest rainwater for landscape use | R1 *: 0%, R2: 25% R3: 50% R4: 75% R5: 100% households with implemented systems | 5 | 25% |
| M: Incorporate smart meters that detect excessive consumption | M1 *: 0%, M2: 25% M3: 50% M4: 75% M5: 100% households with smart meters | 5 | 25% |

Total Combinations: 125. * Indicative of a base value.

Since interventions R and M require a rainfall parameter, we rely on historical observation data from the Parameter elevation regression on the independent Slopes model (PRISM) which map the daily precipitations as a function of the temperature, rain shadows, and orography. Prism data are estimated for the Boerne city at a resolution of the 800-m. To extract the trend of the historical precipitations, we used the Seasonal and Trend decomposition using Loess (STL) method [39] and obtained the long-term change which includes the linear trend and the interannual to decadal variability. The long-term trend is then trained using Bayesian state space modeling by assuming t-distributions of the trend to account for the heavy tails (e.g., extremes) in the long-term component, and the slopes and the intercepts are assumed to follow a random walk process. We then generated a posterior distribution of 10,000 samples using a Humiliation Markov Chain Monte Carlo (MCMC) sampling method [40,41].

The inference period is from 1980 to 2019, where we broke down the daily precipitation into the four seasons, adjusted to match the shoulder months used in modeling demand. The forecasting periods is 51 yrs from 2020 to 2070. Finally, the mean of posterior distribution is then used as input for the model. Using statistical model-based predictions of historical and future precipitation is preferred over the global climate model to constrain the uncertainty in historical and future change in climate driven trend. Specifically, the MCMC sampling allows us to generate the whole distributions of possible change in the climate-driven trend instead of a single realization that could be obtained from the climate model. Also, the PRISM data take into account the local climate changes at the city scale (~800 m), which could be beyond the capability of the climate models to simulate. Although precipitation was modeled based on historical data, the extent, severity, and timing of future hydrologic and meteorological droughts are unpredictable.

In addition to the top interventions, we also modeled reductions in indoor residential consumption as a standalone strategy. Indoor residential per capita consumption was reduced from 0.35 $m^3$ (92 gallons) to 0.25 $m^3$ (67 gallons) in 0.02 $m^3$ (5 gallon) increments. This increment was chosen given its interpretability and visualization by residents as well as for the greater feasibility of small reductions. For reference, Stave [10] models the effects of a single 25-gallon reduction in indoor demand in Las Vegas, Nevada, from 0.29 $m^3$ to 0.19 $m^3$ (76 to 51 gallons) PCD.

## 3. Results

Parameter variation was analyzed under a base scenario and two drought scenarios (Table 4). The base scenario assumed that total maximum supply of *potable* water for the city

would be available throughout the simulation period. The effects of hydrologic droughts are less easily understood because these impact processes that can take months or years to manifest. Therefore, we consider only a meteorological drought by decreasing forecasted precipitation by 50%, which was experienced in the region during the 2011, one-year drought of record. As mandated by the drought contingency plan of the city, total rate of consumption (indoor plus outdoor) is expected to decrease to 0.45 m$^3$ (120 gallons) PCD, an approximate 22.5% reduction from the 0.59 m$^3$ (155 gallons) PCD base rate. Drought scenario (DS) one (DS1) assumes that the entire reduction in demand is met by reducing outdoor consumption. Drought scenario two (DS2), on the other hand, equally distributes the expected reduction between indoor and outdoor.

**Table 4.** Description of Scenarios.

| Scenario | Description |
|---|---|
| Base Scenario | Supply is firm throughout the simulation period, i.e., maximum * total supply is available to the city. Demand is calculated using base values. |
| Drought Scenario 1 | Supply is firm throughout the simulation period, i.e., maximum * total supply is available to the city. For demand projections, outdoor demand is reduced to meet the decreased total consumption for the city of 120 GPCD under drought conditions. |
| Drought Scenario 2 | Supply is firm throughout the simulation period, i.e., maximum * total supply is available to the city. For demand projections, indoor *and* outdoor demand are reduced to meet the decreased total consumption for the city of 120 GPCD under drought conditions. |

* Maximum total supply is equal to the sum of the sources in Table 1.

The following results are organized according to these three scenarios. Each of the listed figures will contain a vertical line in the year 2045 (the current planning period for Boerne), and horizontal lines indicating Total (for total water supply), a 10% threshold (10% below total water supply) and Current Supply Behavior (current water supply at 50% of contracted availability from GBRA). Furthermore, only 25 out of the 125 combinations generated different results in each scenario, but the listed figures only include the five most salient combinations since the same interpretation is achieved and figures are more legible.

*3.1. Comparisons to Base Scenario*

Figure 5a shows the model output for the base scenario. The base combination of strategies, $X_1R_1M_1$, is the projected water consumption based on normal—and current—per capita consumption patterns. From Figure 5a, it is apparent that intervention R (rainwater harvesting) is the dominant variable influencing water demand. Interventions X and M (xeriscaping and smart metering, respectively) appear to influence water demand to a much lesser extent. When intervention X increases from 0% to 5%, demand output decreases by 1.2%. Results indicate that this inverse relationship is directly proportional in nature. When adoption of intervention R increases from 0% to 25%, the demand decreases by approximately 5.5%; similar to intervention X, reductions in demand are linearly related to the adoption rate for intervention R. No adoption of intervention M leads to a measurable decrease in demand.

By 2046, the city will need to rely more heavily on surface water supplies regardless of the combination of interventions employed (Figure 5a); water demand by 2070, under base case conditions, nearly surpasses the 10% threshold. It is clear that water demand without any interventions will monotonically increase with time, just given the projected increase in community population. In the base case scenario, demand surpasses current supply in 2026. If the community adopts rainwater harvesting as an intervention strategy, this crossover point can be delayed; results show that increasing adoption from 25% to 100% (in 25% increments) delays the crossover until 2029, 2030, 2033, and 2046 (respectively, depending on the adoption level). All combinations are below Total water supply.

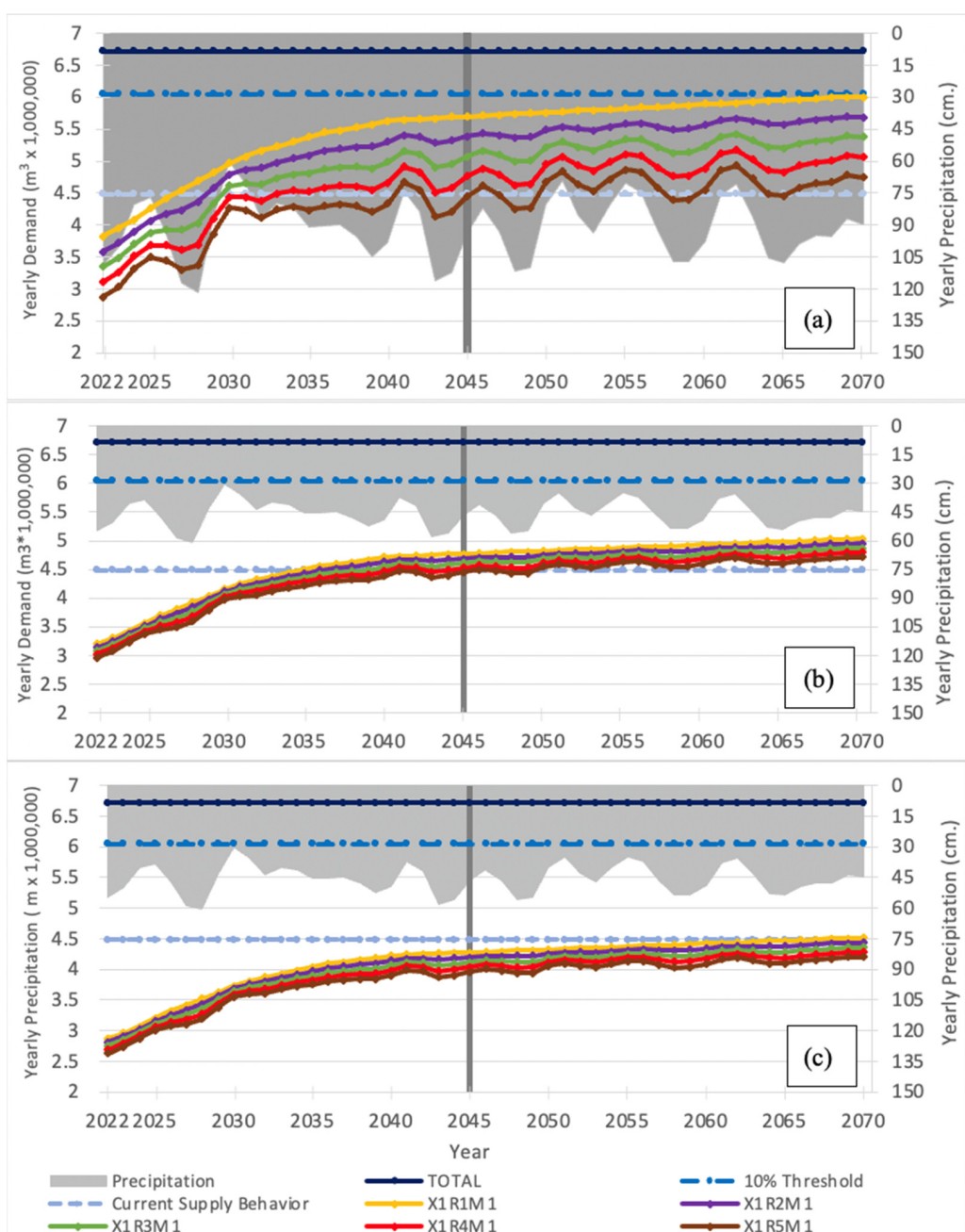

**Figure 5.** Impact of Interventions Based on Drought Scenarios. (**a**) Base case, (**b**) Drought scenario one, (**c**) Drought scenario two.

### 3.2. Drought Scenario One (DSI)

While a 50% reduction in precipitation for the entire simulation period is unlikely to occur, this scenario allows us to study the magnitude of the effects of a meteorological drought at different points in time. Results show that drought conditions manifest a distinct difference from base case conditions (Figure 5b). The spread of the five different combinations shows a smaller relative influence of intervention R (rainwater harvesting), because there is less precipitation available to capture and use as an alternative water supply for irrigation. Thus, during times of drought, when additional water is needed most, rainwater harvesting is less effective. Similar to base case conditions, ranging combinations by interventions X and M have minimal impact. Increasing intervention X from 0% to 5% decreases demand by 0.7%, and intervention M does not reduce demand noticeably at any

level of adoption. Increasing adoption of intervention R from 0% to 25% decreases demand by 1.5%; this reduction is less than that seen in the base scenario.

Under these drought conditions, and assuming outdoor demand is decreased to meet the goal of 0.45 m$^3$ (120 gallons) PCD, the current supply will not be sufficient in the coming years. Base demand will surpass current supply by 2035. Increasing the percentage of residential homes using rainwater harvesting forestalls this crossover point. Increasing adoption from 25% to 100% initially forestalls exceedance of supply from 2036 to 2046, using the modeled precipitation forecast. Increasing adoption to 75% and 100% leads to different behavior; namely, even with these high adoption rates, demand surpasses current supply in 2041 and 2046, respectively, but demand eventually falls below current supply for a few years as a result of higher forecasted precipitation, leading to higher water capture. This outcome is an artifact of the precipitation forecast. Eventually, after years with lower precipitation, water demand again exceeds current supply in 2045 and 2050 for 75% and 100% adoption rates, respectively. Water demand in Boerne is not anticipated to surpass Total supply by 2070 under all demand projections.

*3.3. Drought Scenario Two (DS2)*

Similar to DS1, the mandated reduction in total demand does improve the flexibility of the city to manage drought. In this scenario, a 22% reduction is applied to both indoor and outdoor water use, resulting in approximately 0.08 m$^3$ (20 gallons) per capita reduction in indoor and 0.05 m$^3$ (14 gallons) per capita reduction in outdoor (Figure 5c). While it remains that an increasing population increases the difficulty of managing demand during a dry period, DS2 shows that the city is better equipped when both indoor and outdoor demands are decreased. Results indicate that only under base case conditions does demand surpass current supply, and not until 2068. With all other combinations of interventions X, R, and M demand remains below current supply throughout the model period. When intervention X increases from 0% to 5% demand decreases by 1.2%, while increasing adoption of intervention R from 0% to 25% decreases demand by 1.7%. Intervention M was not effective in reducing demand at any level of adoption.

*3.4. Decreasing Indoor Residential Demand*

While not a strategy chosen by community members, decreasing indoor residential water demand could be important for altering the balance between demand and supply. This scenario tests reductions in indoor residential consumption independent of interventions X, R, and M, and precipitation rates. With the baseline rate of 0.35 m$^2$ (92 GPCD), demand exceeds current supply in 2026. Reducing baseline indoor daily demand in 0.02 m$^3$ (5 gallon) increments alters this balance (Figure 6) but initially only by an additional year, e.g., demand exceeds current supply in 2027 when demand is reduced from 0.35 m$^3$ to 0.33 m$^3$ (92 to 87 gallons) PCD. Further reduction to 0.27 m$^3$ (72 gallons) PCD delays the deficit until 2032, and 0.25 m$^3$ (67 gallons) PCD delays the deficit until 2035. With further 0.02 m$^3$ (5 gallon) decreases in indoor PCD use, each increment will delay a deficit by larger time increments.

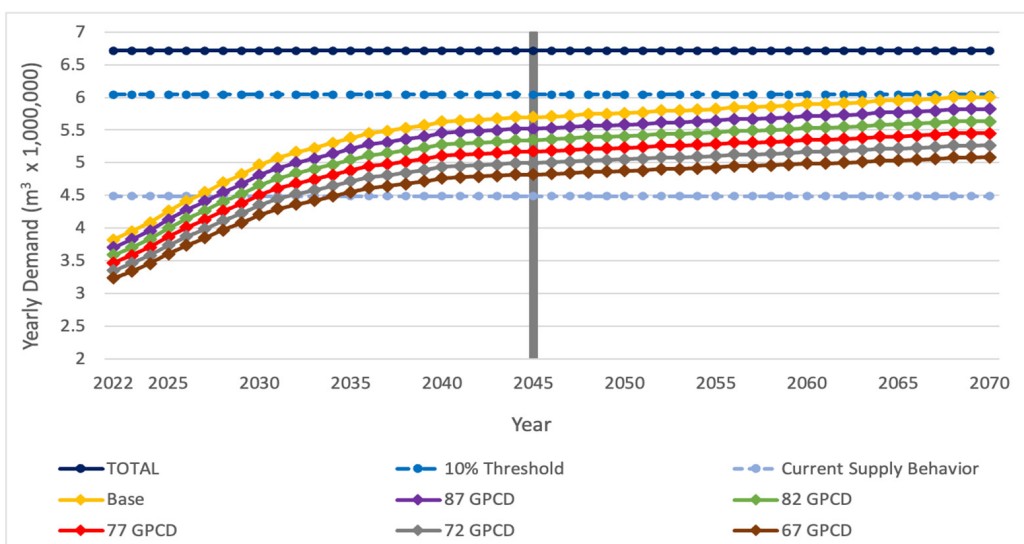

**Figure 6.** Reductions to Residential Indoor Consumption.

## 4. Discussion

The results of this research highlight three important points. First, optionality resulting from multiple combinations of interventions that yield the same or similar results allows the city to accomplish the same goal while accommodating for economic, political, or environmental constraints. For example, combinations $X_2R_1M_1$, $X_2R_1M_2$, $X_2R_1M_3$, $X_2R_1M_4$, and $X_2R_1M_5$ all yield the same demand. In this case, the city could realize 5% water potential savings from xeriscaping, no adoption of rainwater catchment or meters and yield the same results as 100% adoption of meters. From the standpoint of feasibility, understanding sensitivities of the supply/demand balance to changes in interventions can inform policy makers where requests of residents can yield the highest impact and value.

These tradeoffs are important given that different constraints almost always come at some cost, monetary and otherwise. As exemplified by the above example, adopting meters can be a costly undertaking; thus, there is a benefit to communities by achieving similar results without asking residents to sacrifice convenience or cost. The same logic can apply to combinations of interventions that yield different outputs, but where the cost per unit decrease in water savings may be too high to justify the cost of intervention. As shown in Section 3.2, choosing $X_2$ instead of $X_1$ can yield a demand that is 1.2% lower. While any decrease in potable water demand has positive effects, the cost of more households altering their lawns for a 1.2% decrease may not be cost effective. Assessments such as this allow communities to consider and agree to optimal choices.

The second important point is the susceptibility to negative impacts from decreased precipitation, depending on the type of intervention. Precipitation rates can affect both demand and supply. During dry periods, less water for irrigation is supplied naturally (through rainfall) or through rainwater harvesting, just when more water is needed, increasing demand for potable sources. As both drought scenarios indicate, the ability to conserve potable water through adoption of rainwater harvesting or smart meters depends on precipitation. Therefore, with a lack of precipitation, rainwater harvesting and smart meters fail as water-shortage adaptations. Although the model used herein showed little relationship between smart meter adoption and decreased demand, this finding is also based on projected precipitation. Actual precipitation patterns could lead to different effects that may impact the ability of water supply to meet demand.

A further note on precipitation is that, while this study highlights the year in which the total maximum supply will be insufficient to meet its demand under different circumstances, rainfall patterns can actually affect crossover points on smaller temporal scales, such as daily or seasonal. Figure 4 shows that precipitation has the capacity to reduce demand below what is expected. It stands to reason, therefore, that the absence of precipi-

tation can increase demand beyond what is expected. This could occur within a dry, hot summer where demand exceeds previously known peak summer demands and strains water production capacity on a daily basis. In these instances, storage infrastructure such as tanks or impoundments could be effective in restoring system balance.

Finally, while the maximum potable supply, 10% threshold, and current supply intake are shown in Figure 5a–c, the effects of precipitation on overall water sources should be considered. For example, under prolonged drought, decreased spring flow or water table elevation could decrease recharge into Canyon Reservoir and Boerne Lake, as well as (vertical) groundwater recharge. By understanding the magnitude and time necessary to see effects on the physics of the overall hydrologic cycle, the "true" supply availability in a hydrologic drought is determined.

By using SD modeling to reveal the relationships (reinforcing or mitigating) between different parameters in the system, the city is better able to assess the cascading effects of different strategies. For example, reducing indoor residential consumption effectively reduces the amount of potable water used, while at the same time reducing the supply of treated wastewater. Also, increasing the capacity of wastewater treatment plants increases the supply of treated wastewater, but it also reduces the amount of water diverted for Cibolo Creek streamflow, potentially affecting the health of that waterway. Lastly, while rainwater harvesting systems are conservation strategies that can have a positive impact on potable consumption, their conservation benefit depends how much and how often precipitation occurs.

## 5. Conclusions

This work sought to leverage local knowledge, through focus group sessions and a survey, to create a simulation model of the water system in Boerne, and to use such a model to understand the effectiveness of water sustainability and resilience efforts. Stakeholder participation throughout the modelling process ensured that the final product was capable of meeting city needs. Results show that the city can effectively reduce its total demand from potable sources, forestalling when a 10% threshold from maximum supply will be surpassed through various means (Figure 5a–c). While different strategy combinations can achieve the same goal, the effects of climate change can diminish flexibility, especially as population continues to grow. This research has generated a series of options that can be compared and chosen, given local political, economic, environmental, or technological constraints.

This study also highlights the importance of conservation strategies, targeting high demand, increased use of treated wastewater, and education and participation of citizens in water management and planning. In the case where new cost-effective sources become available, conservation remains the best strategy from cost-competitiveness [42] and sustainability standpoints. Furthermore, by treating the root of the problem (i.e., high demand), demand for potable sources and reliance on precipitation can decrease. This is important given the popularity of a strategy such as rainwater harvesting, the effectiveness of which is highly dependent on precipitation events. Finally, Brown [43] argues that awareness of the public and their ability to *see* water stress can strengthen the political viability of conservation strategies. By improving how water-related information is communicated to residents and other stakeholders, the status of these water sources can become visible to the public, and hence more understood.

Although a case study of Boerne, the findings are also relevant in other areas where increasing competition for water sources, future water shortages, and an increasingly warmer and drier climate threaten the sustainability and resilience of water systems. This study could serve as a template for other communities in the Texas Hill Country, or elsewhere, who are interested in leveraging stakeholder feedback for water shortage mitigation and could benefit from conservation strategies, increased use of alternative sources such as treated wastewater, and increased education of citizens. As shown by this study, community feedback and knowledge can inform modeling efforts used to

analyze interventions to reduce demand. In this manner, solutions are based on community feedback and needs, which is inherently a local- or municipal-scale approach.

**Supplementary Materials:** The following supporting information can be downloaded at: https://www.mdpi.com/article/10.3390/w14223682/s1.

**Author Contributions:** Conceptualization, V.R. and M.H.Y.; methodology, V.R., M.H.Y., A.R., K.F. and B.D.L.; software, V.R.; validation, V.R., M.H.Y. and A.R.; formal analysis V.R.; investigation, V.R.; resources, V.R.; data curation, V.R.; writing—original draft preparation, V.R.; writing—review and editing, V.R., M.H.Y., K.F., A.R. and B.D.L.; visualization, V.R.; supervision, M.H.Y.; project administration, M.H.Y.; funding acquisition, V.R. and M.H.Y. All authors have read and agreed to the published version of the manuscript.

**Funding:** Funding by the Gates Millennial Fellowship for V.R. for her graduate research assistantship is gratefully acknowledged. We appreciate the funding to M.H.Y. and the related Internet-of-Water program by the Cynthia and George Mitchell Foundation, grants G-2006-56988 and G-2109-57741.

**Data Availability Statement:** Some or all data, models, or code that support the findings of this study are available from the corresponding author upon reasonable request. Some or all data, models, or code generated or used during the study are proprietary or confidential in nature and may only be provided with restrictions.

**Acknowledgments:** We are indebted to Ben Eldredge and Margaret Lamar of the Cibolo Center for Conservation in Boerne, Michael Mann of the City of Boerne Utilities Department, and Ashley Ward of Duke University's IoW program. We are grateful for the comments of the anonymous reviewers, who improved the manuscript.

**Conflicts of Interest:** The authors declare no conflict of interest.

## Appendix A. Community Engagement Outcomes

Two methods were used for community engagement, a series of focus group sessions and a survey. As mentioned in Section 2.3, these are a part of a greater Internet of Water (2021) project developing in Boerne. Through late 2020 and early 2021, the Internet of Water team from Duke University partnered with the Cibolo Center for Conservation and the University of Texas at Austin to conduct four focus group sessions around municipal leaders, environmental advocates, business leaders, and representatives of socially vulnerable groups. These groups were chosen in order to represent varying perspectives and opinions from community members, an approach that has been previously used by the Internet of Water team from Duke University in other states. Through these sessions, groups discussed water concerns, data needs, and informed the causal loop diagram used for this study. Key takeaways of these focus group sessions, as highlighted by [33] included the desire for consistency across agencies for the collection of data, a need for visuals such as tables and charts for better data interpretation, and increased transparency in data management.

In March 2021, the Internet of Water team from Duke University once again partnered with the Cibolo Center for Conservation and the University of Texas at Austin to develop and distribute a survey through the Office of the Mayor in Boerne, Texas. During a two-week period [33] Boerne residents were able to complete the survey, which generated a total of 324 responses, of which the majority self-identified as residential water consumers; other possible identities included commercial water customer, municipal leader, member of the business community, member of the agricultural community, or environmental advocate. The survey was meant to highlight citizen opinions with respect to water, trusted sources of data and information, concerns over water, and strategies citizens could be willing to support and employ. Of most importance to this specific research was the question regarding strategies citizens would be willing to employ to reduce their consumption of water. This question asked participants to choose three strategies from a list of 10 options: xeriscaping; rainwater harvesting for outdoor use; smart meters; rainwater harvesting for potable use; drip irrigation for landscaping; adopt greywater reuse; upgrade to water efficient fixtures; install raingardens to filter stormwater and recharge groundwater; add

hot water circulator to reduce water waste; and use pool covers to reduce evaporation. The most referenced strategies were the ones used in this research: rainwater harvesting, xeriscaping, and smart meters.

## Appendix B. Statistical Metrics for Model Verification and Validation

As part of Section 2.5, individual Summer demand values were increased to assess statistical error metrics. As Table A1 shows, and as mentioned in Section 2.5, these metrics become worse off as Summer demand values are increased.

**Table A1.** Error Metrics while varying outdoor per capita demand values for summer season observations.

| Year | | 0.24 m³PCD (63 GPCD) | 0.25 m³PCD (65 GPCD) | 0.26 m³PCD (70 GPCD) | 0.28 m³PCD (75 GPCD) | 0.3 m³PCD (80 GPCD) | 0.32 m³PCD (85 GPCD) | 0.34 m³PCD (90 GPCD) |
|---|---|---|---|---|---|---|---|---|
| Summer 2016 | N | 136 | 136 | 136 | 136 | 136 | 136 | 136 |
| | MAD or MAE | 1806.546 | 1874.455 | 2082.361 | 2332.448 | 2620.574 | 2947.978 | 3303.766 |
| | MSE | $1.29 \times 10^9$ | $1.36 \times 10^9$ | $1.61 \times 10^9$ | $1.98 \times 10^9$ | $2.47 \times 10^9$ | $3.07 \times 10^9$ | $3.80 \times 10^9$ |
| | RMSE | 2210.724 | 2268.054 | 2469.395 | 2738.204 | 3056.599 | 3410.733 | 3790.756 |
| | MBE | (586.977) | (775.746) | (1247.809) | (1719.872) | (2191.879) | (2663.887) | (3135.977) |
| | MAPE | 20.137 | 21.122 | 23.931 | 27.112 | 30.610 | 34.430 | 38.471 |
| | NSE | (0.068) | (0.124) | (0.333) | (0.639) | (1.042) | (1.543) | (2.141) |
| Summer 2017 | N | 153 | 153 | 153 | 153 | 153 | 153 | 153 |
| | MAD or MAE | 1740.941 | 1706.793 | 1683.509 | 1718.729 | 1833.681 | 2066.819 | 2416.992 |
| | MSE | $1.05 \times 10^9$ | $1.02 \times 10^9$ | $1.05 \times 10^9$ | $1.20 \times 10^9$ | $1.48 \times 10^9$ | $1.90 \times 10^9$ | $2.44 \times 10^9$ |
| | RMSE | 1991.338 | 1965.892 | 1989.185 | 2130.510 | 2368.991 | 2678.910 | 3038.298 |
| | MBE | 352.684 | 154.532 | (340.813) | (836.207) | (1331.502) | (1826.871) | (2322.413) |
| | MAPE | 15.593 | 15.564 | 16.029 | 16.987 | 18.581 | 21.086 | 24.463 |
| | NSE | (0.028) | (0.001) | (0.025) | (0.176) | (0.454) | (0.860) | (1.392) |
| Summer 2018 | N | 153 | 153 | 153 | 153 | 153 | 153 | 153 |
| | MAD or MAE | 2448.799 | 2408.734 | 2334.805 | 2300.252 | 2355.495 | 2496.773 | 2724.091 |
| | MSE | $1.95 \times 10^9$ | $1.93 \times 10^9$ | $1.99 \times 10^9$ | $2.19 \times 10^9$ | $2.54 \times 10^9$ | $3.02 \times 10^9$ | $3.65 \times 10^9$ |
| | RMSE | 2716.681 | 2704.911 | 2745.173 | 2880.505 | 3098.469 | 3382.983 | 3718.928 |
| | MBE | 267.022 | 58.924 | (461.582) | (981.940) | (1502.397) | (2022.656) | (2543.137) |
| | MAPE | 22.477 | 22.529 | 22.875 | 23.534 | 24.878 | 26.857 | 29.447 |
| | NSE | (0.042) | (0.033) | (0.064) | (0.171) | (0.355) | (0.615) | (0.952) |
| Summer 2019 | N | 153 | 153 | 153 | 153 | 153 | 153 | 153 |
| | MAD or MAE | 2301.190 | 2258.339 | 2197.523 | 2197.091 | 2290.931 | 2487.852 | 2771.511 |
| | MSE | $1.84 \times 10^9$ | $1.79 \times 10^9$ | $1.75 \times 10^9$ | $1.88 \times 10^9$ | $2.17 \times 10^9$ | $2.61 \times 10^9$ | $3.21 \times 10^9$ |
| | RMSE | 2640.396 | 2599.659 | 2577.267 | 2669.103 | 2864.181 | 3143.526 | 3486.735 |
| | MBE | 588.156 | 369.493 | (177.313) | (723.971) | (1270.802) | (1817.509) | (2364.266) |
| | MAPE | 18.365 | 18.350 | 18.676 | 19.460 | 20.921 | 23.103 | 25.866 |
| | NSE | (0.014) | 0.017 | 0.034 | (0.036) | (0.193) | (0.437) | (0.768) |
| Summer 2020 | N | 153 | 153 | 153 | 153 | 153 | 153 | 153 |
| | MAD or MAE | 2301.190 | 2258.339 | 2197.523 | 2197.091 | 2290.931 | 2487.852 | 2771.511 |
| | MSE | $1.84 \times 10^9$ | $1.79 \times 10^9$ | $1.75 \times 10^9$ | $1.88 \times 10^9$ | $2.17 \times 10^9$ | $2.61 \times 10^9$ | $3.21 \times 10^9$ |
| | RMSE | 2640.396 | 2599.659 | 2577.267 | 2669.103 | 2864.181 | 3143.526 | 3486.735 |
| | MBE | 588.156 | 369.493 | (177.313) | (723.971) | (1270.802) | (1817.509) | (2364.266) |
| | MAPE | 18.365 | 18.350 | 18.676 | 19.460 | 20.921 | 23.103 | 25.866 |
| | NSE | (0.014) | 0.017 | 0.034 | (0.036) | (0.193) | (0.437) | (0.768) |

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
