# Peer review of "System Dynamics Modeling in Local Water Management: Assessing Strategies for the City of Boerne, Texas"

_water, doi:10.3390/w14223682_

Round 1

Reviewer 1 Report

Review of System Dynamics Modeling in Local Water Management: assessing strategies for the City of Boerne, Texas

I really appreciated the manuscript that is well structured and organized. In order to improv the quality of the manuscript I have two major comments/suggestions:

1)     At line 127 you write that the chosen approach is tailored on local knowledge and direct feedback of utility managers and stakeholders. This is for sure a strength point of your research, but it could be also a weakness point if you do not stress how your work is useful for other towns. So I would suggest to stress the importance of sharing your experience

2)     At lines 214-217 I understand that you perform annual water balance, assuming the all the annual water supply volume is available since 1st January. In some part of your manuscript you recognize that a daily description of water supply could be more performant. My question is: why haven’t you performed a daily description of water supply?

Minor comments

I think that on a international journal the International system for quantities is enough and appropriate, I would remove Gallons, AF etc…

Simulations are from 2016 to 2070 (lines 230-234) but calibration is from 2016-2020. In my opinion it is a little bit strange that the same period (2016-2020) is used for calibration and simulation. Can you clarify?

Actually, I don’t know what is a shoulder month. Is there another term to identify “shoulder months”?

Line 274: if 50% of the total indoor demand goes to the wastewater treatment, why the remaining 50% of the total indoor demand does not go to the wastewater treatment? It has a large negative environmental impact.

Lines 302-303: it is not clear which variable was considered for calibration. From Table 2 I argue it is a volume, but it is not clear which volume of water: residential? Indoor?

Table 2b: what is Base+2….(+27) etc. for?

Line 333: what is PCD for? Maybe it was written somewhere, but I missed.

Lines 336-344 are not clear form

Line 389 which precipitation is decreased by 50%. Precipitation forecasted according to climate change or historical precipitation?

Table 4: in the description is used the term Maximum: I don’t understand respect to what scenarios is maximum

Line 413 and 415 I don’ understand why the relationship is linear. Maybe you mean “directly proportional”? It is different.

Author Response

Review of System Dynamics Modeling in Local Water Management: assessing strategies for the City of Boerne, Texas

I really appreciated the manuscript that is well structured and organized. In order to improv the quality of the manuscript I have two major comments/suggestions:

1. At line 127 you write that the chosen approach is tailored on local knowledge and direct feedback of utility managers and stakeholders. This is for sure a strength point of your research, but it could be also a weakness point if you do not stress how your work is useful for other towns. So I would suggest to stress the importance of sharing your experience

  • Response: We appreciate the reviewer’s comment and agree that more can be written about the importance of stakeholder engagement. To build on other statements about community feedback throughout the manuscript, we added content in the concluding remarks of the manuscript. See line 769 and lines 774-777.

2. At lines 214-217 I understand that you perform annual water balance, assuming the all the annual water supply volume is available since 1st In some part of your manuscript you recognize that a daily description of water supply could be more performant. My question is: why haven’t you performed a daily description of water supply?

  • Response: The reviewer raises an important point. First, we note that a goal of the model—to assess broader benefits of specific interventions on the long-term supply/demand dynamics of this community—were better achieved and more illustrative by pooling a year’s worth of supply at the beginning of the year, rather than using daily supply dynamics. Trying to simulate the dynamics of community member decision-making on their daily water consumption (e.g., will they irrigate the day after precipitation?) would over-reach the abilities of the model, making calculations and extrapolations for an approximately 50-year time period difficult and likely ambiguous. We believe that the seasonal approach as articulated in Section 2.5 was appropriate given our study goals. We have included this explanation to reduce ambiguity in the manuscript. See lines 222-248.

Minor comments

I think that on a international journal the International system for quantities is enough and appropriate, I would remove Gallons, AF etc…

  • Response: We have considered the reviewer’s comment regarding units. To maintain benefits for a wider number of interested readers, especially those in the US where the study was conducted, we believe it is best to include English units in parentheses. We hope the Editors will agree, but we are fine with their removal.

Simulations are from 2016 to 2070 (lines 230-234) but calibration is from 2016-2020. In my opinion it is a little bit strange that the same period (2016-2020) is used for calibration and simulation. Can you clarify?

  • Response: We agree with the reviewer’s comment and agree we were unclear on these simulation periods. The calibration period, shown in Figure 4, is from June 01, 2016 through February 26, 2021, which spanned the period for which all water-source data were available. We have revised the caption of Figure 4 to reflect this more clearly. As shown by Figure 5, the simulation period is from January 1, 2022 through the end of year 2070. The model simulation begins on January 1 because this is when the yearly water supply enters the system. See line 263, lines 356-357, and Figure 4.

Actually, I don’t know what is a shoulder month. Is there another term to identify “shoulder months”?

  • Response: We appreciate the question. A shoulder month is a colloquial term for the transition period between seasons. We revised the statements to be more clear. See lines 370-372.

Line 274: if 50% of the total indoor demand goes to the wastewater treatment, why the remaining 50% of the total indoor demand does not go to the wastewater treatment? It has a large negative environmental impact.

  • Response: The 50% of indoor demand going to wastewater treatment plants was chosen based on local knowledge and design specifications used by system operators. Data on wastewater treatment from Boerne does not distinguish water source, whether wastewater or stormwater. Therefore, we followed the city’s guidance in choosing this percentage. See lines 439-440.   

Lines 302-303: it is not clear which variable was considered for calibration. From Table 2 I argue it is a volume, but it is not clear which volume of water: residential? Indoor?

  • Response: We appreciate the reviewer bringing this confusion to the forefront. Total demand includes demand across all sectors, and for indoor and outdoor use. We included this clarification in the manuscript. See line 461.

Table 2b: what is Base+2….(+27) etc. for?

  • Response: We recognize that these column names may be confusing to the reader and appreciate the reviewer’s comment for pointing that out. Overall, the table represents how various error metrics change according to the different residential outdoor per capita consumption values. The columns represent increases in residential outdoor per capita consumption, above the base case of 63 GPCD. We have changed the column names so that their meaning is understandable. See Table 2b and B1.

Line 333: what is PCD for? Maybe it was written somewhere, but I missed.

  • Response: Line 55 explains that PCD stands for per capita per day. This has been restated for the benefit of readers. See line 257.

Lines 336-344 are not clear form

  • Response: We would like to address this comment, but we are unsure of what the reviewer is referencing.

Line 389 which precipitation is decreased by 50%. Precipitation forecasted according to climate change or historical precipitation?

  • Response: Forecasted precipitation was decreased by 50% to simulate a meteorological drought, as was experienced during the 2011 one-year drought of record. We have made this more explicit in the manuscript. See Line 578.

Table 4: in the description is used the term Maximum: I don’t understand respect to what scenarios is maximum

  • Response: We acknowledge that this may be confusing to the reader. We explain maximum total supply as a footnote in Table 4.

Line 413 and 415 I don’ understand why the relationship is linear. Maybe you mean “directly proportional”? It is different.

  • Response: We agree with the reviewer’s comment that the relationship is directly proportional and have made the change in the manuscript. See line 607.

Reviewer 2 Report

See attached comments.

Author Response

Review comments – “System Dynamics Modeling . . . Boerne, Texas” Reviewer: Jim F. Chamberlain

Overall: Clear and well-written. Will be of interest to many readers.

Specific:

Line 177: Is water reclamation really a conservation strategy? Seems like it is an increase in supply rather than reduction in demand. Just re-consider.

  • Response: We appreciate the reviewer’s comment. This specific question was discussed at length during early stages of the research. After various conversations with the Utilities Director, we decided that water reclamation was best classified as a conservation strategy. We reasoned that wastewater reclamation is not an increase in supply that can be used for indoor consumption. Rather, its use reduces pressure on potable water supply that would otherwise be used for outdoor irrigation. Therefore, increased use of reclaimed water represents a conservation strategy for reducing potable water use. We have included this explanation in the manuscript. See lines 179-185.

258: Can this 80/20 breakdown be adjusted in the model? Might be quite different in other cities / towns.

  • Response: Yes, the assumed 80/20 breakdown between residential and non-residential water-use can be adjusted in the model. This assumption was necessary to differentiate the different sources of demand given that the city did not record this. Other cities may have demand data specific to each sector or they may need to make educated assumptions, as was done here. We included a sentence in the manuscript that indicates this. See lines 367-368.

Table 3. Increment for Xeriscaping – should this read as 5% increment?

  • Response: Yes, the increment for xeriscaping is 5%. We appreciate the reviewer noticing this mistake. The increment has been changed to reflect a 5%. See Table 3.

346: The method of referencing in the document changes from [ ] format to a “Author (date)” format. Please be consistent throughout the document.

  • Response: We appreciate the reviewer’s noticing this mistake. We have reviewed the manuscript and made the changes necessary.

627: “....from a list of 11 options”. I think many readers would be consider to know what options were given. Please list these in this section.

  • Response: We appreciate the comment and have included a run-in list of the options given. See lines 830-834.
